# Comprehensive mutagenesis to identify amino acid residues contributing to the difference in thermostability between two originally thermostable ancestral proteins

**Satoshi Akanuma**[1]*, **Minako Yamaguchi**[2], **Akihiko Yamagishi**[2]

**1** Faculty of Human Sciences, Waseda University, Tokorozawa, Saitama, Japan, **2** Department of Applied Life Sciences, Tokyo University of Pharmacy and Life Sciences, Hachioji, Tokyo, Japan

* akanuma@waseda.jp

**Data Availability Statement:** All relevant data are within the manuscript and its Supporting Information files.

## Abstract

Further improvement of the thermostability of inherently thermostable proteins is an attractive challenge because more thermostable proteins are industrially more useful and serve as better scaffolds for protein engineering. To establish guidelines that can be applied for the rational design of hyperthermostable proteins, we compared the amino acid sequences of two ancestral nucleoside diphosphate kinases, Arc1 and Bac1, reconstructed in our previous study. Although Bac1 is a thermostable protein whose unfolding temperature is around 100°C, Arc1 is much more thermostable with an unfolding temperature of 114°C. However, only 12 out of 139 amino acids are different between the two sequences. In this study, one or a combination of amino acid(s) in Bac1 was/were substituted by a residue(s) found in Arc1 at the same position(s). The best mutant, which contained three amino acid substitutions (S108D, G116A and L120P substitutions), showed an unfolding temperature more than 10°C higher than that of Bac1. Furthermore, a combination of the other nine amino acid substitutions also led to improved thermostability of Bac1, although the effects of individual substitutions were small. Therefore, not only the sum of the contributions of individual amino acids, but also the synergistic effects of multiple amino acids are deeply involved in the stability of a hyperthermostable protein. Such insights will be helpful for future rational design of hyperthermostable proteins.

## Introduction

Elucidation of the mechanisms underlying protein thermostability is an important issue not only for understanding the amino acid sequence-structure-stability relationship, but also for developing proteins that can be used in industrial processes [1–4]. Comparative structural analyses between thermophilic and mesophilic protein structures have provided insights into the molecular mechanisms responsible for the thermostability of thermophilic proteins [5,6]. The mechanisms suggested by those studies include the formation of intra- or inter-molecular ion-pairs and

**Funding:** This work was supported by JSPS KAKENHI (Grant Number 19K21903) and Individual Research Allowance of Waseda University to SA, and Basic Research Fund of Tokyo University of Pharmacy and Life Sciences to AY. The funders had no role in study design, data collection and analysis, decision to publish, or preparation of the manuscript. There was no additional external funding received for this study.

**Competing interests:** The authors have declared that no competing interests exist.

ion-pair networks in the native structure [7–9] and also in the denatured state [10], increased hydrophobicity and improved packing in the hydrophobic core [11–14], shorter loops [15,16], improved subunit-subunit interactions [3,17,18] and entropic advantages due to the increased flexibility of native structures [19]. However, we have yet to reach a comprehensive understanding of the mechanism of protein thermostability [20]. For example, one paper reported that no consistent trends were found between the amino acid composition of proteins and their stabilities [16], while another paper pointed out that small non-polar amino acids are more frequent in thermostable proteins [21]. Furthermore, in many cases, these factors would only partially explain the differences in thermostability between thermophilic proteins and their mesophilic homologues. A major hindrance in identifying all of the amino acid residues involved in determining the differences in thermostability between a pair of homologous proteins is the fact that many mutations that do not affect thermal stability have accumulated during evolution.

Previously, we inferred ancestral amino acid sequences of nucleoside diphosphate kinases (NDKs) possessed by the last archaeal common ancestor and the last bacterial common ancestor [22]. NDK is an enzyme that catalyzes the transfer of γ-phosphate of a nucleoside triphosphate to a nucleoside diphosphate. We built phylogenetic trees by comparing the amino acid sequences of NDKs possessed by extant archaea and bacteria, and reconstructed several archaeal and bacterial common ancestral NDKs. Among the reconstructed NDKs, the archaeal common ancestor Arc1 and the bacterial common ancestor Bac1 exist as homo-hexamers with a protomer of 139 amino acid residues, and only 12 amino acid residues differ between their amino acid sequences (Fig 1A). The unfolding midpoint temperature of Arc1 (113°C) is 14°C higher than that (99°C) of Bac1 at pH 7.0. Crystal structures of Arc1 and Bac1 have been reported (PDB codes: 3VVT and 3VVU). Ramachandran plots computed by the PROCHECK module of SAVES (https://saves.mbi.ucla.edu/) showed that only one residue (Val113) existed in disallowed regions in both Arc1 and Bac1 structures (S1 Fig). Superposition of the structure of Arc1 upon that of Bac1 (Fig 1B) yielded a root mean square deviation of alpha carbons ($C_\alpha$ rmsd) of 0.61 Å for the 139 aligned residues, as calculated using PDBeFold ver. 2.59 (https://www.ebi.ac.uk/msd-srv/ssm/). Reduced non-polar accessible surface areas, and increased numbers of inter-subunit ion pairs and hydrogen bonds were proposed as structural features that likely contribute to the high thermal stabilities of Arc1 and Bac1 based on the comparison of their crystal structures with that of the NDK from the mesophilic *Dictyostelium discoideum* (see reference [22] for more details).

In this study, in order to understand the structural mechanisms by which originally very thermostable proteins become more stable, we first created twelve Bac1 mutants in which one amino acid was replaced by the residue found at the same position in the amino acid sequence of Arc1. Three mutants indeed showed an enhanced unfolding midpoint temperature of greater than 2°C at pH 6.0, pH 7.0 and pH 7.6. Moreover, combinations of the three beneficial amino acid substitutions further improved the thermal stability of Bac1, thus producing a mutant that displayed an unfolding midpoint temperature quite similar to that of Arc1 at pH 7.6. Therefore, only three of the 12 amino acids that differ between Arc1 and Bac1 play a major role in the difference in the thermostabilities of the two proteins. However, the other nine amino acids in Arc1 contribute somewhat to its greater thermostability although their individual contributions are small. Finally, we compared the tertiary structures of Arc1 and Bac1 and discuss the structural mechanisms for the enhanced stability of the Bac mutant bearing the three amino acid substitutions.

## Materials and methods

### Site-directed mutagenesis

The gene encoding the ancestral NDK corresponding to the last bacterial common ancestor had been cloned into pET21c [22]. Site-directed mutagenesis was carried out on the Bac1 gene

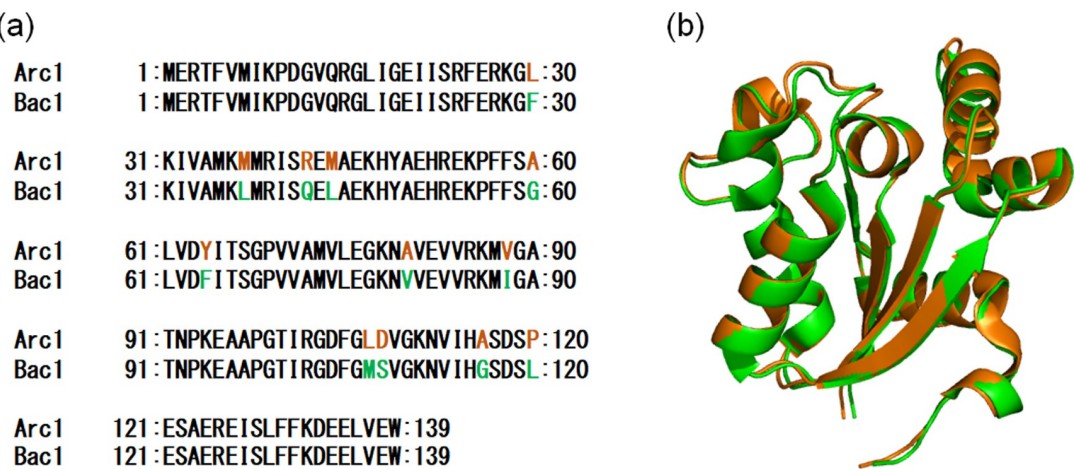

**Fig 1. Comparison of the amino acid sequences and tertiary structures of the ancestral NDKs.** (a) Pair-wise amino acid sequence alignment of Arc1 and Bac1. Inconsistent amino acid residues are colored. (b) Superimposed structures of Arc1 (orange) and Bac1 (green). Superimposition was performed with PDBeFold ver. 2.59 (https://www.ebi.ac.uk/msd-srv/ssm/) and the superimposed structures were visualized with PyMOL (https://pymol.org/2/).

using the splicing-by-overlap extension (SOE) PCR method [23]. SOE PCR consists of two rounds of PCR using the outer primer pairs (T7_forward: 5'- TAATACGACTCACTATAGG-3' and T7_reverse: 5'-GCTAGTTATTGCTCAGCGG-3') that are common to all reactions and the inner mutagenic primer pairs (S1 Table). A pair of mutagenic primers have complementary sequences at their 3′-ends. T7_forward and a mutagenic reverse primer were used for amplification of the 5′ gene segment (PCR #1). A mutagenic forward primer and T7_reverse primer were used for amplification of the 3′ gene segment (PCR #2). Both amplifications used pET21c carrying the gene encoding Bac1 as the template. The two amplified fragments generated in PCR #1 and PCR #2 were purified using the FastGene Gel/PCR Extraction Kit (Nippon Genetics) and then mixed. They were used for PCR amplification again in the presence of T7_forward and T7_reverse primers (PCR #3). All PCR amplifications were performed in a reaction mixture containing 1× KOD -plus- PCR buffer (Toyobo), 1 mM $MgSO_4$, 0.2 mM each of dNTPs, 25 µM each of a pair of primers, and 1.0 unit of KOD -plus- DNA polymerase. The time-temperature program was as follows: step 1, 95˚C, 3 min; step 2, 95˚C, 30 s; step 3, 55˚C, 30 s; step 4, 68˚C, 1 min. Steps 2–4 were repeated 25 times. After PCR #3, the product was again purified using the FastGene Gel/PCR Extraction Kit (Nippon Genetics).

## Construction of expression plasmids

The amplified DNA encoding each Bac1 mutant was digested with *Nde*I and *Bam*HI (New England Biolabs) and then purified by agarose gel electrophoresis. The purified DNA was ligated into the *Nde*I-*Bam*HI site of pET21c and then used to transform *Escherichia coli* JM109. Plasmid DNA was prepared from each transformant after cultivating overnight in 2 mL of Luria-Bertani (LB) medium supplemented with 150 µg/ml of ampicillin at 37˚C. The resulting plasmid DNA was subjected to DNA sequencing to confirm the nucleotide sequence of the region encoding the Bac1 mutant.

## Protein preparation

For preparation of the Bac1 mutants, *E. coli* Rosetta2 (DE3) was transformed with each of the expression plasmids. Each transformant was cultured overnight at 37˚C in 2 L of LB medium

supplemented with 150 μg/ml of ampicillin. Overexpression was induced by the Overnight Express Autoinduction System (Novagen-Merck). The cells were then harvested, resuspended in 20 mM Tris, pH 7.5, 1 mM EDTA, and disrupted by sonication. The cell lysates were centrifuged at 60,000 x g for 20 minutes. The resulting supernatant was individually heat-treated at 75°C for 20 min and then centrifuged again at 60,000 x g for 20 min. To purify the protein, the supernatant was filtered and then subjected to HiTrapQ (Cytiva) column chromatography. The adsorption buffer was 20 mM Tris, pH 7.5, 1 mM EDTA, and the elution buffer was 20 mM Tris, pH 7.5, 1 mM EDTA, 1 M NaCl. The fraction containing the Bac1 mutant was collected and dialyzed overnight against 20 mM Tris, pH 8.8, 1 mM EDTA, followed by ResourceQ (Cytiva) column chromatography. The adsorption buffer was 20 mM Tris, pH 8.8, 1 mM EDTA, and the elution buffer was 20 mM Tris, pH 8.8, 1 mM EDTA, 1 M NaCl. The fractions that were homogeneous judging from the results of Coomassie blue staining after SDS-polyacrylamide gel electrophoresis were used for subsequent analysis.

### Quantification of protein concentration

Protein concentrations were determined using the $A_{280}$ values of the protein solutions. The molar absorption coefficient at 280 nm for each Bac1 mutant was calculated as reported by Pace and colleagues [24], who modified the procedure described by Gill and von Hippel [25].

### Thermal stability measurement

The $T_m$ value of each protein was estimated from the thermal denaturation curve obtained by monitoring the change in ellipticity at 222 nm using a J-1100 spectropolarimeter (Jasco) equipped with a programmable temperature controller and a pressure-proof cell compartment that prevented the aqueous solution from bubbling and evaporating at high temperatures. Protein solutions were diluted to a final concentration of 20 μM with 20 mM potassium phosphate (pH 6.0 or 7.6), 50 mM KCl, 1 mM EDTA. A 0.1 cm path-length cell was used. The temperature was increased at a rate of 1.0°C/min.

### Activity measurement

The enzymatic reaction catalyzed by Arc1, Bac1 and its mutants was assayed at 70°C by monitoring the increase in the amount of the product ATP using the Kinase-Glo Plus Luminescent Kinase Assay kit (Promega). The assay solution consisted of 50 mM HEPES (pH 8.0), 25 mM KCl, 10 mM $(NH_4)_2SO_4$, 2.0 mM $Mg(CH_3COO)_2$, 1.0 mM dithiothreitol, 1.0 mM ADP and 2.5 mM GTP. One enzyme unit equaled 1 μmol ATP formed per min. The Michaelis constant values ($K_m$s) for the substrate ADP and the catalytic rate constant ($k_{cat}$) were calculated based on the steady-state kinetic data with an assay solution of 50 mM HEPES (pH 8.0), 25 mM KCl, 10 mM $(NH_4)_2SO_4$, 2.0 mM $Mg(CH_3COO)_2$, 1.0 mM dithiothreitol, 2.5 mM GTP, with ADP at concentrations between 50 and 1000 μM. The kinetic parameters were calculated by nonlinear least-square fitting of the steady-state velocity data to the Michaelis-Menten equation using the Enzyme Kinetics module of SigmaPlot Ver. 13 (Systat Software).

## Results

### Mutagenesis of residues in the bacterial common ancestral NDK, Bac1

A pairwise alignment of the amino acid sequences of Arc1 and Bac1 (Fig 1A) showed that 127 out of 139 aligned residues are identical between the two proteins. Therefore, only 12 sites were occupied by different amino acids in the two proteins. To identify amino acid substitutions that enhance the thermal stability of Bac1, an amino acid in Bac1 was substituted with

the residue found at the same position in Arc1. Accordingly, we created twelve mutants of Bac1.

The genes encoding the mutant proteins were PCR amplified and then overexpressed in *E. coli* Rosetta2 (DE3). Each mutant protein was purified to homogeneity by successive column chromatography steps and then used for temperature-induced unfolding experiments (S2 Fig). The unfolding experiments were first performed at pH 6.0, pH 7.0 and pH 8.0. Although all of the proteins showed cooperative two-state transitions at pH 6.0 and pH 7.0, G60A and S108D showed atypical unfolding curves at pH 8.0 (S3 Fig). In particular, S108D seemed to undergo a structural change in which the content of helical structure increases, rather than denaturing into a random coil, as the temperature rose. Therefore, the thermal unfolding experiments were also performed at pH 7.6. The unfolding midpoint temperatures ($T_m$) of Arc1, Bac1 and the 12 mutants are listed in S2 Table and compared in Fig 2. Three mutants (S108D, G116A, L120P) had significantly increased $T_m$s of greater than 2˚C compared to Bac1 at pH 6.0, pH 7.0 and pH 7.6. The mutant with the highest $T_m$ value among the mutants was G116A, which exhibited 5–7˚C greater $T_m$s than those of Bac1. The $T_m$ value of S108D was 6˚C greater than that of Bac1 at pH 7.6, whereas the $T_m$s of the mutant was only 3˚C greater than those of Bac1 at pH 6.0 and pH 7.0. Therefore, the negative charge of the aspartate residue's side chain is thought to be responsible for the increase in thermostability of S108D. The $T_m$ values of L120P were also higher than those of Bac1 by 2–3˚C between pH 6.0 and pH 7.6. Conversely, no mutants showed decreased $T_m$s of greater than 2˚C between pH 6.0 and 7.6. At pH 8.0, four mutants (L37M, Q42R, L44M, I88V) showed decreased $T_m$s compared to Bac1.

To understand the effects of amino acid substitutions on catalytic properties, the kinetic parameters of Arc1, Bac1 and the 12 mutants were obtained from steady-state kinetic experiments using various concentrations of ADP and 2.5 mM GTP (Fig 3 and S2 Table). The specific activity measurements of Arc1 and Bac1 as a function of temperature revealed that both ancestral NDKs showed the highest specific activity at 70˚C under the condition employed (S4 Fig). Therefore, it is reasonable to expect that all of the Bac1 mutants also showed the highest specific activity at 70˚C. Accordingly, the steady-state kinetic experiments were performed at 70˚C. It should be noted that the twelve mutated positions are not in the active site. The hexamers of Arc1 and Bac1 have six active sites, respectively, and each active site consists of residues in a single subunit. In addition, the inter-subunit contacts are robust even at unfolding temperature (see Discussion). Therefore, the effect of subunit dissociation on activity was not considered. To ensure high affinity with the substrate at a very high temperature, hyperthermophilic enzymes should require smaller $K_m$ values at a given temperature than those of their less stable homologues. However, Arc1 had a $K_m$ value for ADP (430 μM) that is 2.4 times more unfavorable than that (180 μM) of Bac1. The overall enzyme efficiency is customarily expressed as $k_{cat}/K_m$. The $k_{cat}/K_m$ value (4.1 μM$^{-1}$ s$^{-1}$) of Arc1 was two times better than that of Bac1 (2.0 μM$^{-1}$ s$^{-1}$) due to the $k_{cat}$ value (1800 s$^{-1}$) of Arc1 that was 4.9 times greater than that (370 s$^{-1}$) of Bac1. Six mutants (L37M, Q42R, L44M, M107L, G116A, L120P) had smaller $k_{cat}/K_m$ values compared to Bac1. Conversely, F30L, G60A, F64Y, V80A, I88V and S108D had greater $k_{cat}/K_m$ values compared to Bac1. When compared to the kinetic parameters of Arc1, all of the Bac1 mutants showed more favorable $K_m$ values but smaller $k_{cat}$ and $k_{cat}/K_m$ values.

## Combination of the beneficial mutations

Fig 2 shows that no single mutation was sufficient to improve the thermal stability of Bac1 to the same level as that of Arc1. It is reasonable to assume that combining multiple beneficial

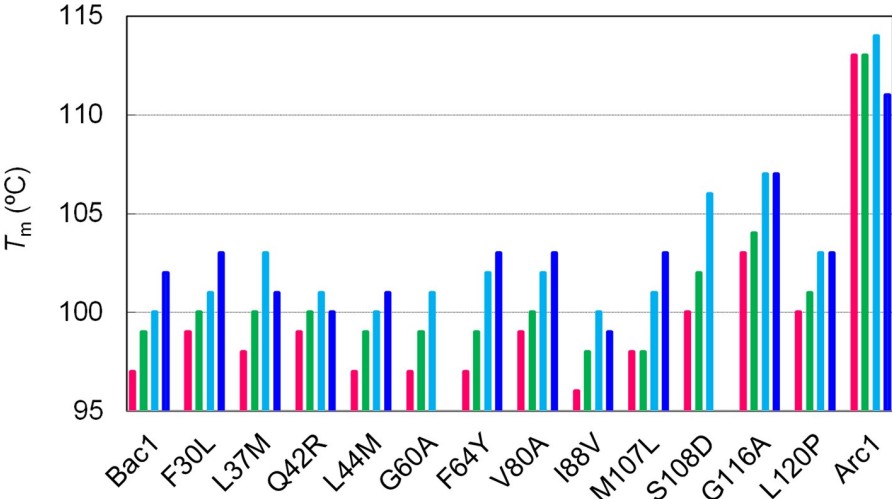

**Fig 2. $T_m$s of Arc1, Bac1 and its mutants at pH 6.0 (magenta), pH 7.0 (green), pH 7.6 (cyan) and pH 8.0 (blue).**
The $T_m$ values were estimated from the data shown in S2 Fig. $T_m$ values are not presented for G60A, S108D, S108D/G116A and S108D/G116A/L120P at pH 8.0 because atypical unfolding curves were observed for the proteins at pH 8.0 (S3 Fig).

substitutions in a protein would further improve its thermostability if the effects of the substitutions are not conflicting. Therefore, we simultaneously introduced S108D and G116A substitutions into Bac1. Temperature-induced unfolding curves of the resulting double mutant (S108D/G116A) at pH 6.0, pH 7.0 and pH 7.6 were compared to those of Arc1, Bac1 and G116A (Fig 4). Similar to the S108D mutant, S108D/G116A showed atypical unfolding curves at pH 8.0. The $T_m$ value (105˚C) of S108D/G116A at pH 6.0 was 8˚C and 2˚C higher than those of Bac1 and G116A, respectively, but 8˚C lower than that of Arc1. At pH 7.0, $T_m$ (107˚C) of S108D/G116A was 8˚C and 3˚C higher than those of Bac1 and G116A, respectively, and 6˚C lower than that of Arc1. At pH 7.6, the $T_m$ (112˚C) of S108D/G116A was 12˚C and 5˚C higher than those of Bac1 and G116A, respectively, and only 2˚C lower than that of Arc1. Thus, a greater synergistic effect of the combination of the two amino acid substitutions was observed at pH 7.6 than at pH 6.0 and pH 7.0.

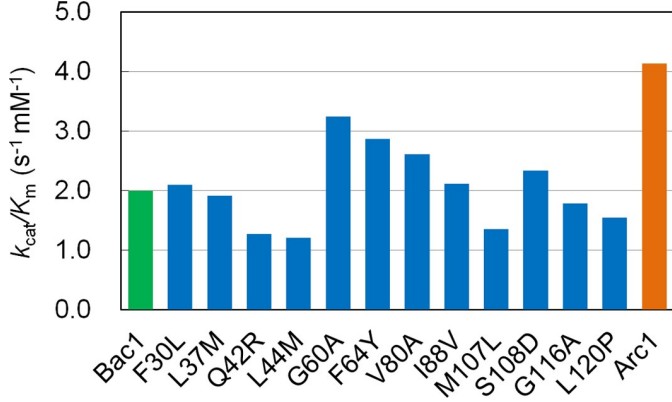

**Fig 3. The $k_{cat}/K_m$ values of Arc1, Bac1 and its mutants at 70˚C.** $K_m$ for ADP, and $k_{cat}$ were calculated by nonlinear least-square fitting of the steady-state kinetic data to the Michaelis-Menten equation using the Enzyme Kinetics module of SigmaPlot Ver. 13 (Systat Software, Richmond) and listed in S2 Table.

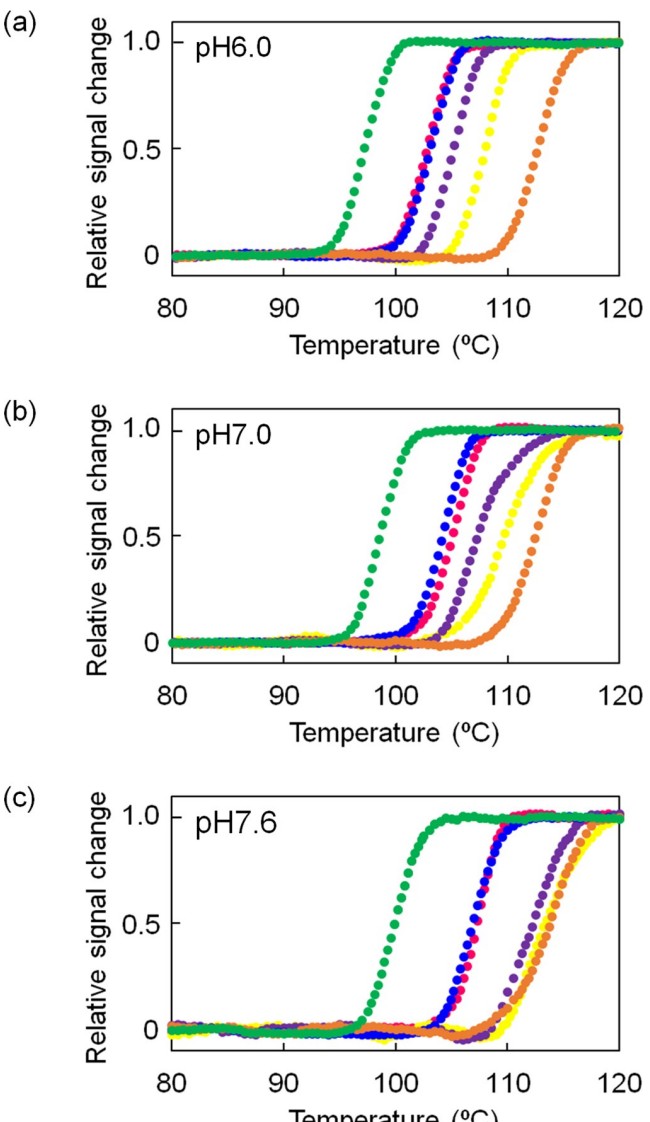

**Fig 4.** Temperature-induced unfolding curves of Arc1, Bac1 and its mutants at pH 6.0 (a), pH 7.0 (b) and pH 7.6 (c). The change in ellipticity at 222 nm was monitored as a function of temperature. The scan rate was 1.0˚C/min. The samples contained 20 μM protein in 20 mM potassium phosphate (pH 6.0 or 7.6), 50 mM KCl, 0.5 mM EDTA. The plots were normalized with respect to the baseline of the native and denatured states. Orange, Arc1; green, Bac1; blue, G116A; purple, S108D/G116A; yellow, S108D/G116A/L120P; magenta, Bac1-mu9.

We also introduced the L120P substitution into the S108D/G116A mutant because the substitution also increased the $T_m$ of Bac1 by 2–3˚C. Fig 4 shows that the resulting S108D/G116A/L120P mutant had further improved thermostability. Moreover, the $T_m$ value (113˚C) of the resulting S108D/G116A/L120P mutant was close to that (114˚C) of Arc1 at pH 7.6. Thus, although twelve amino acids are different between the sequences of Arc1 and Bac1, the difference in $T_m$ of 14˚C between the two ancestral NDKs can be almost eliminated at pH 7.6 by only three amino acid substitutions.

The kinetic parameters of S108D/G116A and S108D/G116A/L120P at 70˚C are provided in S2 Table. The $K_m$ values of the double and triple mutants were between those of Bac1 and Arc1. The $k_{cat}$ values of both mutants were slightly greater than that of Bac1 but much smaller

than that of Arc1. The mutants had $k_{cat}/K_m$ values that are smaller than that of Bac1. Thus, unlike thermostability, the catalytic efficiency was not improved by the combination of the two or three amino acid substitutions.

## Combination of S108D or G116A substitution with other substitutions

As mentioned above, the S108D and G116A substitutions had the most positive impact on the thermostability of Bac1. It is reasonable to expect that multiple amino acid substitutions at sites that are close to each other are more likely to show a synergistic effect than those that are far apart. In the tertiary structures of Bac1 and Arc1, the amino acid residue at position 108 is located at the subunit boundary (Fig 5) and is in close proximity to the amino acid at position 30 of a neighboring subunit and the amino acid at position 107 in the same subunit (Fig 6A). We therefore constructed a mutant protein in which two amino acid substitutions (F30L and M107L substitutions) were combined with the S108D substitution. The $T_m$ value of the resulting F30L/M107L/S108D mutant was 4°C higher than that of the S108D mutant at pH 7.0 (S5 Fig). When the F30L substitution was individually introduced into Bac1, its $T_m$ was increased by only 1°C. Moreover, the M107L substitution decreased the $T_m$ of Bac1 by 1°C. Thus, although the individual contributions of the F30L and M107L substitutions to the improvement in thermostability were small or negative, they showed a synergetic effect with the S108D substitution at pH 7.6.

In the structure of Bac1, Gly116 is located on a β-strand and surrounded by residues 84, 88, 101, and 114 (Fig 6B). In Arc1, the side chain of Ala116 is pointed toward the interior core. The same amino acids occupy positions 84, 101 and 114 between the two ancestral proteins. However, an isoleucine occupies position 88 in Bac1, whereas a valine occupies this position in Arc1. We therefore created one more mutant by introducing the I88V substitution into the G116A mutant. However, the resulting I88V/G116A mutant showed the same $T_m$ value as that of G116A at pH 7.0 (S5 Fig). In other words, regardless of whether residue 116 was Gly or Ala, the I88V substitution did not improve the thermostability of the proteins.

## Combinatorial effects of amino acid substitutions whose individual contributions to the thermostability are small

While three of the amino acid substitutions (S108D, G116A and L120P substitutions) substantially enhanced the thermostability of Bac1, the individual effects of the remaining nine amino

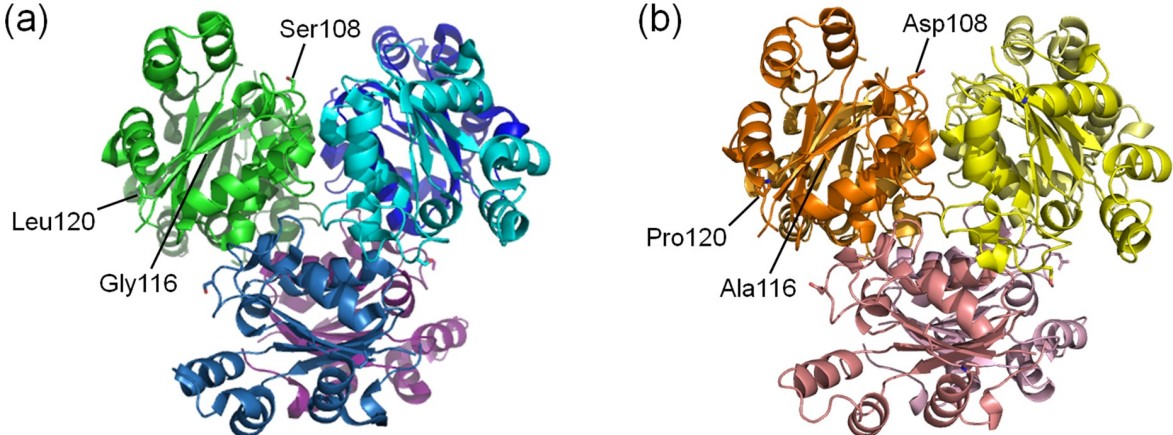

**Fig 5.** Hexameric structures of Bac1 (a) and Arc1 (b). The subunits of the proteins are each colored differently. Residues at positions 108, 116 and 120 in a subunit of each hexamer are indicated.

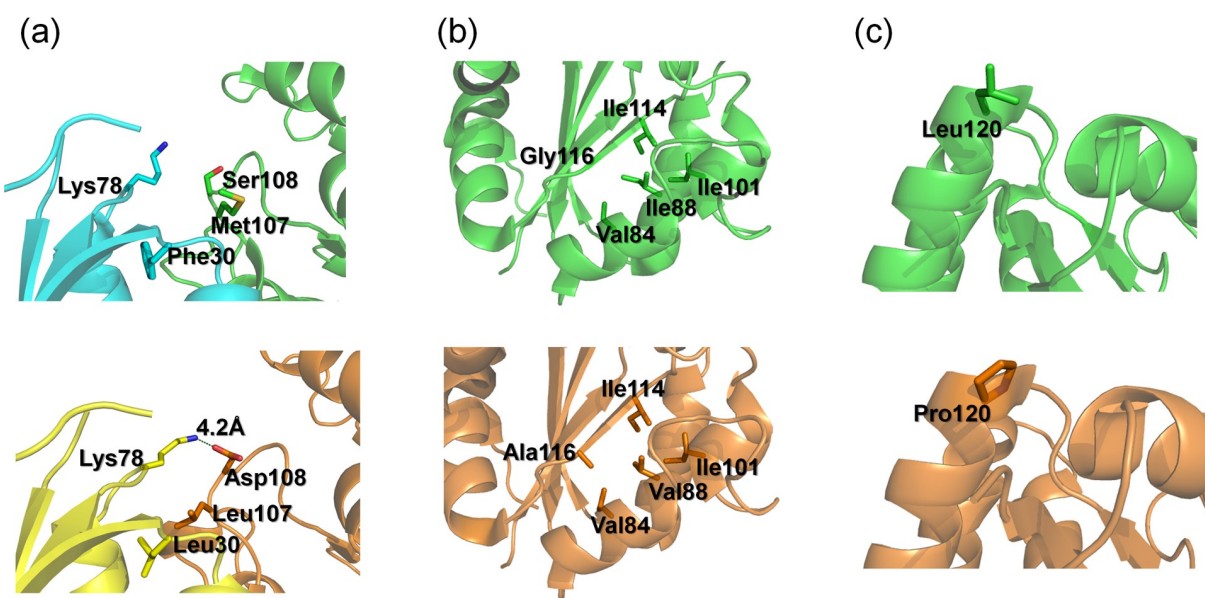

**Fig 6.** (a) Local structures around position 108 of Bac1 (upper panel) and Arc1 (lower panel). Two subunits are colored differently. In Arc1, the carboxyl group of Asp108 forms a middle-range (4.2 Å) ion-pair with the amino group of Lys78 in the adjacent subunit. (b) Structures around position 116. In Bac1 (upper panel), Gly116 is located within a hydrophobic core and is surrounded by aliphatic residues including Val84, Ile88, Ile101 and Ile114. In Arc1 (lower panel), Val and Ala occupy positions 88 and 116, respectively. (c) Leu120 in Bac1 (upper panel) and Pro120 in Arc1 (lower panel) are each located at the N-terminal end of an α-helix.

acid substitutions on the thermostability of Bac1 were small. Therefore, we initially predicted that the remaining nine amino acid substitutions would not improve the thermostability of Bac1 even if the nine amino acid substitutions were introduced into Bac1 together. To test this prediction, we replaced residues 30, 37, 42, 44, 60, 64, 80, 88 and 107 in Bac1 by the amino acids found at the same positions in Arc1. The resulting mutant, named Bac1-mu9, was over-expressed in *E. coli* and then used for temperature-induced unfolding measurements at pH 6.0, pH 7.0, pH 7.6 and pH 8.0 (Fig 4 and S2 Table). Contrary to the expectation, the $T_m$ value of Bac1-mu9 was greater than that of Bac1 by 6°C at pH 6.0 and pH 7.0, by 7°C at pH 7.6, and by 4°C at pH 8.0. At pH 7.0 in particular, the magnitude of the increase in $T_m$ by the combination of amino acid substitutions was greater than the sum of the magnitude of the increase or decrease in $T_m$ by the individual amino acid substitutions.

The kinetic parameters of Bac1-mu9 at 70°C are presented in S2 Table. The $K_m$ value of Bac1-mu9 was more unfavorable than that of Arc1, although the difference is within the margin of error. Bac1-mu9 had a greater $k_{cat}$ value than that of Bac1, but its $k_{cat}/K_m$ value is smaller than that of Bac1.

## Discussion

Several methods for creating more thermostable proteins have been proposed. Two conventional methods to create proteins with enhanced thermostability are rational design [9,14,26,27] and directed evolution [3,28–30]. As alternative approaches, consensus design and ancestral sequence reconstruction (ASR) have also been developed to create thermostable proteins. Consensus design has been used in conjunction with multiple amino acid sequence alignments (MSAs) of homologous proteins [31–33]. Its theoretical basis is that amino acids that contribute to a protein's stability have a higher probability of being selected during the evolutionary process than those that do not contribute to stability [34]. ASR is a way to design

amino acid sequences that are predicted to have been possessed by extinct species [35–39]. In conjunction with the topology of a phylogenetic tree and MSAs of homologous proteins, ASR has also created many thermostable proteins. The design of thermostable proteins by ASR assumes that the ancestral organisms were (hyper)thermophilic and therefore ancestral residues would be responsible for the thermostability of a protein to a much greater extent than non-ancestral residues [40]. Many studies have validated the reliability of consensus design and ASR as methods to create highly thermostable proteins [22,41–48], which could serve as industrially useful tools and good models for examining the structural mechanisms of protein thermostability.

More stable proteins are preferred as potential scaffolds for protein engineering [49,50]. Therefore, it is an attractive challenge to further improve the thermostability of an originally thermostable protein [9,51]. The two ancestral NDKs, Bac1 and Arc1, were created in our previous ASR experiments [22]. The amino acid sequences of the two ancestral proteins are very similar, differing only by 12 out of 139 amino acids. Although Bac1 is a very thermostable protein, Arc1 is much more thermostable and its $T_m$ is greater than that of Bac1 by 14°C at pH 7.0. Therefore, comparison of Bac1 and Arc1 can provide insight into the structural mechanisms by which originally very thermostable proteins become more stable. The difference in thermostability between a pair of homologous proteins has sometimes been discussed on the basis of protein topology and oligomerization state. High-resolution three-dimensional structures of Bac1 and Arc1 have been determined at 2.2 Å resolution by X-ray crystallography [22], showing that both proteins are hexamers composed of six identical subunits each consisting of 139 amino acid residues (Fig 5). Moreover, significant similarity is found between the backbone structures of Bac1 and Arc1 ($C_\alpha$ rmsd = 0.61 Å; Fig 1B). Therefore, differences in protein topology or oligomerization state are not the basis of the differences in thermostability between Bac1 and Arc1.

From the thermostability analysis of the series of mutant proteins, we found that all of the proteins, except for Arc1, showed higher $T_m$s at alkaline pHs than at acidic pH (Fig 2 and S2 Table). Because the isoelectric points of Arc1 and Bac1 are both 6.40, the isoelectric point may not be related to the difference in pH dependence of thermal stability. Perhaps the negative charges of acidic amino acids contribute significantly to the thermostability of Bac1 and its mutants, whereas the positive charges of basic amino acids contribute more significantly to the thermostability of Arc1. Indeed, similar to Arc1, the Q42R mutant showed a lower $T_m$ at pH 8.0 than at pH 7.6, although the extent of the decrease in $T_m$ was smaller than that observed for Arc1. However, the data presented here are not necessarily sufficient to explain the differences in pH-dependent thermal stability. Further experimental and/or computational studies are therefore necessary to explain more precisely the structural features that are related to the pH dependence of thermal stability.

We also found that three (positions 108, 116, 120) out of twelve positions where different amino acids are present in Bac1 and Arc1 are primarily responsible for the difference in thermostability between the two ancestral proteins. Serine108 is located on a surface loop that contacts an adjacent subunit (Figs 5A and 6A). The S108D substitution increased the $T_m$ of Bac1 by 6°C at pH 7.6, but only by 3°C at pH 6.0. The fact that the magnitude of the improvement in thermostability by the S108D substitution is dependent on pH suggests that the negative charge of the aspartate sidechain contributes to the enhanced thermostability. In the Arc1 tertiary structure, the carboxyl group of Asp108 is 4.2 Å away from the positively charged sidechain of Lys78 in a neighboring subunit, forming a middle-range ion pair (Fig 6A). The ion pair is the only possible interaction involving the negative charge of the aspartate side chain. Since the ancestral NDK is a homo-hexamer, the presence of aspartate instead of serine at position 108 generates six more inter-subunit ion pairs. The contribution of an increased number

of inter-subunit ion-pairs to greater protein thermostability has been proposed previously [52,53]. In addition, because residues 78 and 108 are both located in flexible loops, the formation of an ion pair would make the loops more rigid, thus contributing to improved thermostability. Less flexible surface regions have also been pointed out as a key feature associated with great stability of thermophilic proteins [16].

In order to investigate whether the dissociation of subunits occurs during denaturation, we performed denaturation experiments of Bac1 at different protein concentrations. If the subunits are dissociated upon thermal denaturation, then their stability would increase as the protein concentration increases. As shown in S6 Fig, the $T_m$ was 99°C when the protein concentration was 12 μM. In contrast, when the protein concentration was 48 μM, the $T_m$ (98°C) was slightly smaller. Therefore, subunit dissociation associated with degeneration was unlikely to have occurred. Nevertheless, the formation of the inter-subunit ion-pairs in the natural state can enthalpically contribute to the stability of the protein if the ion-pairs are lost in the denatured state.

Another interpretation may be also possible for the molecular mechanism of enhanced thermostability by the S108D substitution. Fig 7 compares the distribution of electrostatic potential in Bac1 and Arc1 hexamers computed by the Adaptive Poisson-Boltzmann Solver (APBS) software [54]. The area around residue 108 is completely positive in Bac1, while the electrostatic potential is shifted toward neutral in Arc1. Therefore, localized charge repulsion may be occurring in Bac1. The presence of a negatively charged aspartate residue at position 108 might mitigate the repulsion between the local positive charges and thus contribute to the greater stability of Arc1 and the S108D mutant compared to Bac1.

The G116A substitution had the greatest impact on the thermostability of Bac1. According to the tertiary structure of Bac1, Gly116 is located on a β-strand and surrounded by the hydrophobic residues Val84, Val88, Ile101 and Ile114 (Fig 6B). It is well accepted that hydrophobicity within a protein's interior core is one of the major driving forces that stabilize the native structures [55]. The contribution of the hydrophobic residues involved in the formation of the hydrophobic core of a hyperthermophilic protein to its extreme thermostability has been suggested by a study using ribonuclease HII from *Thermococcus kodakaraensis* [11]. Arc1 has an alanine instead of a glycine at position 116 (Fig 6B). The additional methyl group of the alanine residue would increase the hydrophobicity of the interior core and thus enhance the thermostabilities of Arc1 and the G116A mutants of Bac1.

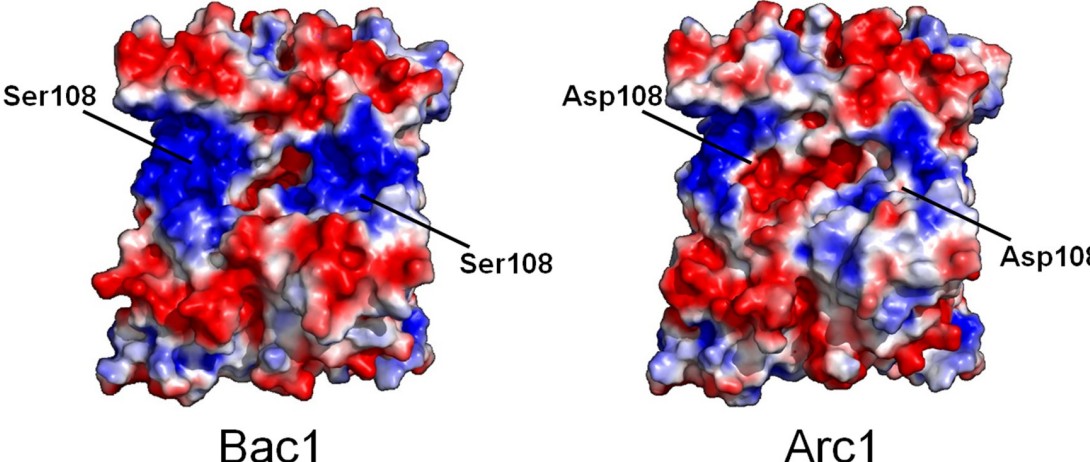

**Fig 7. Electrostatic potentials on the surfaces of Bac1 and Arc1 hexamers.** Electrostatic potentials were calculated and visualized using APBS [54] integrated into PyMOL (https://pymol.org/2/). Red and blue indicate negative and positive potentials, respectively. Ser108 or Asp108 are indicated in each hexameric structure.

However, it cannot simply be expected that an increase in the interior hydrophobicity of a protein improves its thermostability. Because more hydrophobic amino acids have larger side chains, increasing the hydrophobicity of an interior core requires the introduction of larger amino acids, which often causes structural stress. It has been shown that the difference in stability between a pair of proteins can be attributed to both hydrophobic interactions and packing density with an equivalent energetic magnitude [56]. Therefore, improving a protein's stability by increasing the hydrophobicity of the interior core would be successful only when the amino acid substitutions also improve the packing density of the core. The addition of one methyl group to the interior core of Bac1 by the G116A substitution may have been the best way to improve the balance of hydrophobicity and packing density of the core.

Leucine 120 of Bac1 is located at the N-terminus of the sixth α-helix (Fig 6C). Leucine residues have a relatively high helix propensity, whereas prolines have the lowest helix propensity and are considered to be helix breakers [57]. However, it has been pointed out that the introduction of one or a few proline residue(s) into a loop region decreases the flexibility of the denatured state, making the denatured state entropically unfavorable and thus stabilizing the folded structure [58,59]. The greater thermostabilities of Arc1, which has a proline at position 120, and the L120P mutant compared to Bac1 indicate that, as suggested before [60], the presence of a proline at the end of the α-helix can contribute to enhanced protein thermostability.

In this study, we found that three amino acid residues are primarily responsible for the difference in the thermostability between Bac1 and Arc1. In order to explore whether the structural mechanisms for the great thermostability of Arc1 suggested in this study are also used in NDKs from extant hyperthermophiles, we compared the amino acid sequences of extant mesophilic, thermophilic and hyperthermophilic NDKs (S7 Fig). Table 1 shows that all NDKs from hyperthermophiles have an alanine at position 116, while all NDKs from mesophiles have a glycine. Thus, for residue 116, Arc1 has acquired high thermostability by a mechanism that is also employed by the hyperthermophilic NDKs. In contrast, the amino acid residues at positions 108 and 120 do not parallel those found in NDKs from extant hyperthermophiles; there was no tendency for the NDKs from hyperthermophiles to have aspartate and proline at positions 108 and 120, respectively. Rather, a hyperthermophilic NDK from *Sulfurisphaera tokodaii* has a serine at position 108 and some hyperthermophilic NDKs have a positively or negatively charged amino acid residue instead of a proline at position 120. In addition, a mesophilic NDK from *Dictyostelium discoideum* has an aspartate at position 108 and the *Methanothermobacter thermautotrophicus* NDK, whose $T_m$ is 17˚C lower than that of Bac1, has a proline at position 120. Thus, the structural mechanisms that confer high thermostability on Arc1 are not always employed by the hyperthermophilic NDKs found in nature.

## Conclusion

Three out of twelve amino acid residues that differ between Bac1 and Arc1 are likely to be the primary determinants for the difference in the thermostability of the two proteins. Our mutagenesis experiments on Bac1 and comparisons of the tertiary structures around the three positions of Bac1 and Arc1 suggest that the formation of inter-subunit ion pairs, elimination of electronic potential bias, improvement of hydrophobicity and packing effects of the interior core, and introduction of a proline at the terminus of an α-helix contributed to further improving the thermostability of the already thermostable Bac1. However, the remaining nine amino acid residues also contribute to the difference in thermostability between the two ancestral proteins. Therefore, a combination of many factors, some of which have very small individual effects, are responsible for the greater thermostability of Arc1 compared to Bac1. The last point

**Table 1. Optimal growth temperatures (OGT) of eleven microorganisms, $T_m$s of their NDKs and amino acid residues at positions corresponding to those at 108, 116 and 120 in the ancestral NDKs.**

| Source [a] | Accession number [b] | OGT (°C) | $T_m$ (°C) [c] | Amino acid | | |
|---|---|---|---|---|---|---|
| | | | | 108 | 116 | 120 |
| Pho | O58429 | 98 | 111 | E | A | K |
| Ape | Q9Y9B0 | 95 | 108 | D | A | P |
| Mja | Q58661 | 85 | 101 | T | A | E |
| Afu | O29491 | 83 | 100 | D | A | P |
| Sto | Q976A0 | 80 | 107 | S | A | E |
| Tth | Q72GQ0 | 75 | 99 | T | G | L |
| Mth | O26358 | 65 | 82 | E | A | P |
| Tac | Q9HJ59 | 57 | 99 | G | A | P |
| Bsu | P31103 | 37 | 57 | F | G | L |
| Eco | P0A763 | 37 | 56 | S | G | V |
| Ddi | P22887 | 22 | 62 | D | G | V |

[a] The eleven microorganisms are: Pho, *Pyrococcus horikoshii* (strain ATCC 700860/DSM 12428/JCM9974/NBRC 100139/OT-3); Ape, *Aeropyrum pernix* (strain ATCC 700893/DSM11879/JCM 9820/NBRC 100138/K1); Mja, *Methanocaldococcus jannaschii* (strain ATCC 43067/DSM 2661/JAL-1/JCM 10045/NBRC 100440); Afu, *Archaeoglobus fulgidus* (strain ATCC 49558/VC-16/DSM 4304/JCM 9628/NBRC 100126); Sto, *Sulfurisphaera tokodaii* (strain DSM 16993/JCM 10545/NBRC 100140/7); Tth, *Thermus thermophilus* (strain ATCC BAA-163/DSM 7039/HB27); Mth, *Methanothermobacter thermautotrophicus* (strain ATCC 29096/DSM 1053/JCM 10044/ NBRC 100330/Delta H); Tac, *Thermoplasma acidophilum* (strain ATCC 25905/DSM 1728/JCM 9062/NBRC 15155/AMRC-C165); Bsu, *Bacillus subtilis* (strain 168); Eco, *E. coli* (strain K12); Ddi, *Dictyostelium discoideum* (Slime mold).

[b] The accession number for each species obtained from *UniProt* Knowledgebase (https://www.uniprot.org/).

[c] The $T_m$ values of the extant microbial NDKs are from reference 22.

reminds us that not only the sum of the contributions of individual amino acids, but also the synergistic effects of multiple amino acids are deeply involved in the stability of a protein.

## Supporting information

**S1 Fig. Ramachandran plots for the dimeric structures of Arc1 (PDB ID: 3VVT) and Bac1 (PDB ID: 3VVU) computed by the PROCHECK module of SAVES (https://saves.mbi.ucla. edu/).** The most favored regions are represented by red; additional allowed regions are represented by yellow; generously allowed regions are represented by light yellow; disallowed regions are represented by white. Glycine residues are shown as triangles and the other residues are shown as squares. In each plot, only Val113 exists in disallowed regions. (TIF)

**S2 Fig. Temperature-induced unfolding curves of Arc1, Bac1 and its mutants at pH 6.0, pH 7.0, pH 7.6 and pH 8.0.** The change in ellipticity at 222 nm was monitored as a function of temperature. The scan rate was 1.0°C/min. The samples comprised 20 μM protein in 20 mM potassium phosphate, 50 mM KCl, 0.5 mM EDTA. The plots were normalized with respect to the baseline of the native and unfolded states. Orange filled circles, Arc1; green filled circles, Bac1; cyan open circles, F30L; blue open circles, L37M; yellow open circles, Q42R; brown filled circles, L44M; black filled circles, G60A; magenta open circles, F64Y; gray filled circles, V80A; cyan filled circles, I88V; yellow filled circles, M107L; magenta filled circles, S108D; blue, G116A; purple filled circles, L120P. (TIF)

**S3 Fig. Atypical unfolding curves observed for G60A, S108D, S108D/G116A and S108D/ G116A/L120P at pH 8.0.** The raw data for the change in ellipticity at 222 nm are presented as

a function of temperature. The scan rate was 1.0˚C/min. The samples comprised 20 μM protein in 20 mM potassium phosphate (pH 8.0), 50 mM KCl, 0.5 mM EDTA.
(TIF)

**S4 Fig. Temperature dependence of the specific activities of Arc1 and Bac1.** The specific activities were measured using a reaction mixture consisting of 50 mM HEPES (pH 8.0), 25 mM KCl, 10 mM $(NH_4)_2SO_4$, 2.0 mM $Mg(CH_3COO)_2$, 1.0 mM dithiothreitol, 1.0 mM ADP and 2.5 mM GTP. One enzyme unit equaled 1 μmol ATP formed per min. Each value is the average of at least three replicas.
(TIF)

**S5 Fig. Temperature-induced unfolding curves of Bac1 mutants with single or multiple amino acid substitution(s).** The change in ellipticity at 222 nm was monitored as a function of temperature. The scan rate was 1.0˚C/min. The samples comprised 20 μM protein in 20 mM potassium phosphate (pH 7.0), 50 mM KCl, 0.5 mM EDTA. The plots were normalized with respect to the baseline of the native and denatured states. Magenta, S108D; red, F30L/M107L/S108D; blue, G116A; cyan, I88V/G116A.
(TIF)

**S6 Fig. Temperature-induced unfolding curves of Bac1 at two different protein concentrations.** Bac1 was dissolved in 20 mM potassium phosphate, pH 7.0, 50 mM KCl, 1 mM EDTA. The protein concentration was 12 μM (light green) or 48 μM (dark green). The scan rate was 1.0˚C/min. The plots were normalized with respect to the baseline of the native and denatured states.
(TIF)

**S7 Fig. Multiple amino acid sequence alignment of eleven microbial NDKs.** Positions corresponding to those at 108, 116 and 120 of the ancestral NDKs are indicated above the sequences. The sequences were aligned with MAFFT (https://mafft.cbrc.jp/alignment/server/) and visualized using ClustalX (http://www.clustal.org/clustal2/).
(TIF)

**S1 Table. Mutagenic primers used in this study.**
(DOCX)

**S2 Table. The numerical data for $T_m$, $K_m$, $k_{cat}$ and $k_{cat}/K_m$ of Arc1, Bac1 and its mutants.**
(DOCX)

## Author Contributions

**Conceptualization:** Satoshi Akanuma.

**Data curation:** Satoshi Akanuma, Minako Yamaguchi.

**Funding acquisition:** Satoshi Akanuma, Akihiko Yamagishi.

**Investigation:** Satoshi Akanuma, Minako Yamaguchi.

**Project administration:** Satoshi Akanuma, Akihiko Yamagishi.

**Supervision:** Akihiko Yamagishi.

**Writing – original draft:** Satoshi Akanuma.

**Writing – review & editing:** Satoshi Akanuma, Akihiko Yamagishi.

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
