## [Decision Letter · Decision Letter 0]

5 Aug 2021

PONE-D-21-21944

Comprehensive mutagenesis to identify amino acid residues contributing to the difference in thermostability between two originally thermostable ancestral proteins

PLOS ONE

Dear Dr. Akanuma,

Thank you for submitting your manuscript to PLOS ONE. After careful consideration, we feel that it has merit but does not fully meet PLOS ONE’s publication criteria as it currently stands. Therefore, we invite you to submit a revised version of the manuscript that addresses the points raised during the review process.

ACADEMIC EDITOR: Would you like to answer reviewers' questions and try to correct your manuscript according to their serious criticism.

We look forward to receiving your revised manuscript.

Kind regards,

Eugene A. Permyakov, Ph.D., Dr.Sci.

Academic Editor

PLOS ONE

Journal Requirements:

"This work was partly supported by JSPS KAKENHI (Grant Number 19K21903) to SA. The funders had no role in study design, data collection and analysis, decision to publish, or preparation of the manuscript."

Reviewers' comments:

Reviewer's Responses to Questions

Comments to the Author

1. Is the manuscript technically sound, and do the data support the conclusions?

Reviewer #1: Partly

Reviewer #2: Partly

2. Has the statistical analysis been performed appropriately and rigorously?

Reviewer #1: N/A

Reviewer #2: No

3. Have the authors made all data underlying the findings in their manuscript fully available?

Reviewer #1: No

Reviewer #2: No

4. Is the manuscript presented in an intelligible fashion and written in standard English?

Reviewer #1: Yes

Reviewer #2: Yes

5. Review Comments to the Author

Reviewer #1: Comprehensive mutagenesis to identify amino acid residues contributing to the difference in thermostability between two originally thermostable ancestral proteins

The paper explains an effort to modify a thermostable protein Bac1 to hyperthermostable one following some amino acid sequence pattern of one naturally hyperthermostable Arc1 sequence.

No doubt the application of this work is important. But the way the investigation is done is inadequate. The methodology section is insufficient. The database id of different gene or the proteins are missing. The description of the organism, strain etc is not sufficient.

The protocol of this investigation is the several point mutations, but that information like primers and PCR protocol is not there.

Proteins structural validity studies its different energetic value, Ramachandran plot data are missing.

The possible mechanisms of thermostability or hyperstability factors are not clearly discussed.

What is the basis to test only at pH 6 or 7.6? why not 7? If 6 is chosen then why not 8?

Thermal stability increased for all the isolates from pH 6 to pH7.6 except Arc1, add an explanation. I am curious to see one more alkaline pH.

Though hyperthermostable, the Km of Arc1 is very high than Bac1 that suggests its less efficiency.

How the structural superimposition was done?

In fig. S3 not too much change is noticed.

The objective of the work is important and could be useful after these modifications.

Reviewer #2: The authors have presented a comprehensive mutational study on two similar nucleoside diphosphate kinases (NDK), aiming at understanding the role of specific amino acid residues in their extreme thermal stability. They have constructed a comprehensive range of mutant enzymes and studied their thermal stability and enzymatic activity.

The data related to the melting temperature (Tm) of different mutants raises few, if any, questions. However, when the authors turn to enzyme kinetics and its interpretation, they open a whole big can of worms.

For starters, a correct interpretation of the kinetics results requires answering the following questions:

1. How the enzyme activity depends on the oligomeric state of the NDK’s. Is an isolated subunit catalytically active?

2. How the oligomeric state of the enzymes depends on the temperature?

3. What is the temperature optimum for each studied mutant? In the current study the authors have uniformly measured the kinetics at 70C for all. However, if the enzymes’ temperature optimum varies as greatly as Tm, these results cannot be compared in a meaningful way. This point also highlights issues with the experiment planning. I would suggest that the kinetics measurements to be repeated at Topt for each enzyme variant.

Unfortunately, due to the points above, the kinetics part of the study warrants a significant review (probably involving re-planning and re-doing the kinetics). Other remarks are given below.

Major comments

1. What’s the reason to have two pH for the study, 6.0 and 7.6?

2. Ln. 169. Since S108D is one of the important mutations, it would be worth to show a structural fragment of the enzyme with possible interactions for the aspartate.

3. Kinetics discussion: the overall enzyme efficiency is customarily expressed as kcat/Km. Then Fig. 3 could be condensed into a single plate showing bars for kcat/Km only; individual data for kcat and Km must be presented in a table.

4. Ln. 290-291: related to the issues above: in a currently presented form it’s not clear if these intersubunit contacts are important, since the authors don’t know if the melting starts in the aggregated or dissociated protein.

5. The original numerical data for Tm, kcat, Km must be tabulated (at least in supplements). The bar charts should have grid lines for easier referencing. Also, it’s better to combine uniform data on one plot for easier comparison (e.g. Fig. 2 could be combined into a single plate).

6. For the double, triple, mu9 mutants (Fig 4) — it would be beneficial to have measured their kinetics too, at optimal temperature.

Minor and technical comments

7. All non-standard abbreviations must be introduced (e.g. KOD, ln. 89; LB, ln. 100 etc.)

8. Ln. 155-159: This is a standard notation for mutants, no need to dedicate that much space for its description.

9. Ln. 177-178: “better” is not the best word to use in this instance.

10. Ln. 190, 191 and elsewhere: “Ser108�Asp” — I suggest to the authors to stick to uniform standard notation for the substitutions like “S108D”.

11. Ln. 298, 300 — this is “electrostatic potential”, not electronic. Fig. 7 title — a proper reference for APBS must be given, this is a major computational package, not just a “plugin”.

12. Fig. 4 needs a legend.

13. Fig. S1, S2 better be overlaid for easier comparison.

6. PLOS authors have the option to publish the peer review history of their article (what does this mean?). If published, this will include your full peer review and any attached files.

Do you want your identity to be public for this peer review? For information about this choice, including consent withdrawal, please see our Privacy Policy.

Reviewer #1: Yes: Smarajit Maiti

Reviewer #2: No

---

## [Author Response · Author response to Decision Letter 0]

5 Oct 2021

Dr. Eugene A. Permyakov

Academic Editor

PLOS ONE

Dear Dr. Permyakov,

We thank the referees very much for their valuable comments. Basically, we have incorporated all the suggestions from the reviewers into our revised manuscript. The changes that we have made in response to their comments follow with the associated page and line numbers of our revisions in parentheses. In our revised text, the changes are shown in blue.

Responses to Reviewer #1

1) The database id of different gene or the proteins are missing. The description of the organism, strain etc is not sufficient.

In response to the reviewer’s comment, we added the accession numbers and strain names into Table 1. For the two ancestral proteins, their PDB codes were provided in the original manuscript.

2) The protocol of this investigation is the several point mutations, but that information like primers and PCR protocol is not there.

According to the reviewer’s suggestion, the sequences of mutagenic primers used in this study are now listed in S1 Table. The sequences of the universal primers are given in the revised text (p. 5 lines 94 - 95). The protocol for the site-directed mutagenesis was also added to the Materials and methods (p. 5 lines 93 - 103, lines 107 - 108).

3) Proteins structural validity studies its different energetic value, Ramachandran plot data are missing.

According to the reviewer’s suggestion, Ramachandran plots for Arc1 (PDB code: 3vvt) and Bac1 (PDB code: 3vvu) were newly produced and provided as S1 Fig. Related descriptions were added to the Introduction (p. 3 line 65 - p. 4 line 67).

4) The possible mechanisms of thermostability or hyperstability factors are not clearly discussed.

We have already discussed the structural basis for the unusually high thermal stabilities of Arc1 and Bac1 in our previous article (Akanuma et al., PNAS 2013, doi: 10.1073/pnas.1308215110). In that article, we pointed out that reduced non-polar accessible surface areas, and increased numbers of inter-subunit ion pairs and hydrogen bonds likely contribute to their high thermal stabilities. Relevant sentences were added to the Introduction (p. 4 lines 70 - 74).

5) What is the basis to test only at pH 6 or 7.6? why not 7? If 6 is chosen then why not 8?

We first conducted the unfolding experiments at pH 6.0 and pH 8.0. However, some mutants showed atypical unfolding curves at pH 8.0 although all of the proteins showed cooperative two-state transitions at pH 6.0. Therefore, we also conducted the unfolding experiments at pH 7.6. In response to the reviewer’s comment, we provided the unfolding data at pH 8.0 in the revised manuscript. In addition, we conducted new unfolding experiments at pH 7.0, and modified Fig 2, Fig 4, S2 Fig (original S1 and S2 Figs) and S5 Fig (original S3 Fig). We also newly created S3 Fig. Related sentences were added to the revised text (p. 8 lines 174 - 185; p. 9 lines 186 - 189; p. 10 lines 220 -227; p. 11 lines 251 - 257; p. 12 lines 263 - 264; p. 12 lines 277 - 281).

6) Thermal stability increased for all the isolates from pH 6 to pH7.6 except Arc1, add an explanation. I am curious to see one more alkaline pH.

According to the reviewer’s suggestion, we added a discussion that may explain the difference in the pH dependence of thermostability among the proteins (p. 14 lines 321 - 332). As mentioned in response 5), we provided the unfolding data at pH 8.0 in the revised manuscript. 

6) Though hyperthermostable, the Km of Arc1 is very high than Bac1 that suggests its less efficiency.

As Reviewer #1 pointed out, more thermostable enzymes typically have smaller Km values at a given temperature than their less stable homologues. However, as Reviewer #2 commented, the overall enzyme efficiency is customarily expressed as kcat/Km and the kcat/Km value (4.1 µM-1 s-1) of Arc1 was two times better than that of Bac1 (2.0 µM-1 s-1). A related description was added (p. 9 lines 201 - 207).

7) How the structural superimposition was done?

Superposition of the structure of Arc1 upon that of Bac1 and calculation of the root mean square deviation of alpha carbons were performed using PDBeFold ver. 2.59 (https://www.ebi.ac.uk/msd-srv/ssm/). Related sentences are found in the text (p. 4 line 67 - 70) and the legend to Fig. 1b (p. 29 lines 667 - 668).

8) In fig. S3 not too much change is noticed.

Although the original S3 Fig represented the unfolding curves produced at pH 7.6, the newly created S5 Fig (original S3 Fig) represents the unfolding curves at pH 7.0. Related sentences were added to the revised text (p. 11 lines 251 - 257)

Responses to Reviewer #2

9) How the enzyme activity depends on the oligomeric state of the NDK’s. Is an isolated subunit catalytically active?

Since we have not been able to isolate the protein into protomers, the effect of protein hexamerization on enzyme activity has not been fully investigated. However, the active site of each protomer is spatially completely independent, and no allosteric effect was observed. Also, since the NDKs of some organisms (e.g., NDKs from E. coli) exist in a tetrameric form, it seems that hexamerization is not essential for activity. In addition, there was no increase in the denaturation temperature when the protein concentration was increased, suggesting that the inter-subunit interactions are very stable and that dissociation of the hexamer upon thermal denaturation does not occur. Moreover, heat treatment of Arc1, Bac1 and mutants at 100°C for 3 min in the presence of 2% SDS followed by SDS-PAGE revealed a band at a molecular weight corresponding to the hexamer (therefore, we also added 4 M urea to the protein solutions and then heat treated for 10 min prior to SDS-PAGE). Accordingly, the hexameric structures of the ancestral NDKs and the mutants must have been maintained under the activity measurement conditions. Related sentences were added to the Results (p. 9 lines 197 - 201).

10) How the oligomeric state of the enzymes depends on the temperature?

As mentioned above, the thermal stabilities of the proteins are independent of protein concentration. Therefore, the hexameric states are always maintained even at unfolding temperatures. A comparison of the denaturation curves at two different concentrations of Bac1 was provided in S6 Fig. Related sentences were added to the Discussion (p. 15 line 350 - p. 16 line 358).

11) What is the temperature optimum for each studied mutant? In the current study the authors have uniformly measured the kinetics at 70C for all. However, if the enzymes’ temperature optimum varies as greatly as Tm, these results cannot be compared in a meaningful way. This point also highlights issues with the experiment planning. I would suggest that the kinetics measurements to be repeated at Topt for each enzyme variant.

We had known that both Arc1 and Bac1 had the highest specific activity at 70℃ under the conditions employed here. Therefore, it is reasonable to expect that the other Bac1 mutants will also show the highest activity at 70°C under these conditions. Therefore, kinetics analyses were always performed at 70°C. Related sentences were added to the Results (p. 9 lines 192 - 197)

12) What’s the reason to have two pH for the study, 6.0 and 7.6?

Please see our response to comment 5) from Reviewer #1, which addresses this comment. 

13) Ln. 169. Since S108D is one of the important mutations, it would be worth to show a structural fragment of the enzyme with possible interactions for the aspartate.

Fig 6a illustrates the structural fragment of the enzyme with possible interactions for the aspartate. The ion pair between Asp108 and Lys78 in an adjacent subunit is the only possible electrostatic interaction involving the negative charge of the side chain of Asp108 (p. 15 lines 341 - 342).

14) Kinetics discussion: the overall enzyme efficiency is customarily expressed as kcat/Km. Then Fig. 3 could be condensed into a single plate showing bars for kcat/Km only; individual data for kcat and Km must be presented in a table.

According to the reviewer’s suggestion, we modified Fig 3 to show the bars for kcat/Km values. The individual values for kcat and Km are listed in S2 Table. Related sentences were modified (p. 9 line 204 – p. 10 line 211).

15) Ln. 290-291: related to the issues above: in a currently presented form it’s not clear if these intersubunit contacts are important, since the authors don’t know if the melting starts in the aggregated or dissociated protein.

As mentioned above, it is likely that the unfolding started in the hexameric state and dissociation of subunits did not occur upon thermal unfolding. Nevertheless, it is possible that the inter-subunit ion pair between Asp108 and Lys78 in an adjacent subunit contributes to the increased stability of the S108D mutants. Related discussion was added to the revised text (p. 15 line 350 – p. 16 line 358).

16) The original numerical data for Tm, kcat, Km must be tabulated (at least in supplements). The bar charts should have grid lines for easier referencing. Also, it’s better to combine uniform data on one plot for easier comparison (e.g. Fig. 2 could be combined into a single plate).

According to the reviewer’s suggestion, the numerical data for Tm, kcat and Km are tabulated in S2 Table. Grid lines were added to Figs 2 and 3. Fig 2 was modified according to the reviewer’s suggestion.

17) For the double, triple, mu9 mutants (Fig 4) — it would be beneficial to have measured their kinetics too, at optimal temperature.

According to the reviewer’s suggestion, the kinetic parameters for S108D/G116A, S108D/G116A/L120P and Bac1-mu9 were newly determined (S2 Table). Related sentences were added to the revised text (p. 10 line 235 – p. 11 line 240; p. 12 line 282 – p. 13 line 285).

Minor and technical comments

18) All non-standard abbreviations must be introduced (e.g. KOD, ln. 89; LB, ln. 100 etc.)

KOD is the official product name, not an abbreviation. As for LB and ABPS, their full names were added at the first appearance as suggested by the reviewer. 

19) Ln. 155-159: This is a standard notation for mutants, no need to dedicate that much space for its description.

According to the suggestion, the relevant sentences were deleted.

20) Ln. 177-178: “better” is not the best word to use in this instance.

The entire sentence containing the relevant word was modified (p. 9 lines 203 - 204).

21) Ln. 190, 191 and elsewhere: “Ser108�Asp” — I suggest to the authors to stick to uniform standard notation for the substitutions like “S108D”.

As suggested by the reviewer, we changed the notations of amino acid substitutions throughout the manuscript.

22) Ln. 298, 300 — this is “electrostatic potential”, not electronic. Fig. 7 title — a proper reference for APBS must be given, this is a major computational package, not just a “plugin”.

As pointed out by the reviewer, we corrected the phrase “electronic potential” to “electrostatic potential”. In addition, reference 54 was newly cited for APBS (p. 16 line 362). 

23) Fig. 4 needs a legend.

The legend to Fig. 4 is in the original and revised manuscripts (p. 29 lines 680 - 685).

24) Fig. S1, S2 better be overlaid for easier comparison.

S2 Fig (originally S1 and S2 Figs) was modified as suggested.

In addition to the aforementioned changes, we have reconducted the temperature-induced unfolding experiment at pH 6.0 because we noticed that the original data might have been measured at around pH 6.3.

We think that our revised manuscript is much improved and hope that its content satisfies the reviewers and the editor.

Thank you in advance for your favorable consideration. 

Sincerely yours,

Dr. Satoshi Akanuma

Faculty of Human Sciences,

Waseda University,

2-579-15 Mikajima, Tokorozawa,

Saitama 359-1192, Japan.

Phone: +81-4-2946-6727, Fax: +81-4-2947-6811

E-mail: akanuma@waseda.jp

---

## [Editor Report · Decision Letter 1]

6 Oct 2021

Comprehensive mutagenesis to identify amino acid residues contributing to the difference in thermostability between two originally thermostable ancestral proteins

PONE-D-21-21944R1

Dear Dr. Satoshi Akanuma,

We’re pleased to inform you that your manuscript has been judged scientifically suitable for publication and will be formally accepted for publication once it meets all outstanding technical requirements.

Kind regards,

Eugene A. Permyakov, Ph.D., Dr.Sci.

Academic Editor

PLOS ONE
---

## [Editor Report · Acceptance letter]

12 Oct 2021

PONE-D-21-21944R1 

Comprehensive mutagenesis to identify amino acid residues contributing to the difference in thermostability between two originally thermostable ancestral proteins 

Dear Dr. Akanuma:

I'm pleased to inform you that your manuscript has been deemed suitable for publication in PLOS ONE. Congratulations! Your manuscript is now with our production department. 

Kind regards, 

on behalf of

Prof. Eugene A. Permyakov 

Academic Editor

PLOS ONE